# Trends in Prostate Cancer Incidence and Survival by Gleason Score from 2000 to 2020: A Population-Based Study in Northeastern Italy

**DOI:** 10.3390/curroncol32080426

**Published:** 2025-07-29

**Authors:** Martina Taborelli, Diego Serraino, Federica Toffolutti, Ettore Bidoli, Sara De Vidi, Lucia Fratino, Luigino Dal Maso

**Affiliations:** 1Cancer Epidemiology Unit, Centro di Riferimento Oncologico di Aviano (CRO) IRCCS, 33081 Aviano, Italy; serrainod@cro.it (D.S.); federica.toffolutti@cro.it (F.T.); bidolie@cro.it (E.B.); sara.devidi@cro.it (S.D.V.); dalmaso@cro.it (L.D.M.); 2Department of Medical Oncology, Centro di Riferimento Oncologico di Aviano (CRO) IRCCS, 33081 Aviano, Italy; lfratino@cro.it

**Keywords:** Gleason score, incidence trends, Italy, prostate cancer, survival

## Abstract

Prostate cancer trends have evolved over time, mainly due to changes in early detection practices, such as PSA testing. In this study, we looked at long-term trends in prostate cancer incidence and survival in northeastern Italy from 2000 to 2020, with a focus on cancer severity measured by Gleason score. We found that less aggressive cancers (Gleason score 2–6) declined over time, while more aggressive cancers (Gleason score 8–10) remained stable in number, but their proportion among all diagnosed cases progressively increased, especially among men aged 75 and older. Survival has improved across all Gleason score groups, with the most notable gains observed in older patients. These findings highlight the need for age-specific clinical strategies to address the growing burden of aggressive prostate cancer, particularly in older men.

## 1. Introduction

Prostate cancer (PCa) is the second most commonly diagnosed cancer in men worldwide, with over 1.4 million new cases estimated in 2022 [1]. Global differences in screening and detection strategies contribute significantly to variations in incidence and mortality rates, with important implications for early diagnosis and treatment decisions [2]. These differences are particularly pronounced in Europe, with distinct trends observed across countries, mostly shaped by national screening programs [3]. In particular, the widespread adoption of prostate-specific antigen (PSA) testing since the early 1990s has significantly impacted PCa incidence trends, initially leading to a sharp increase in diagnoses, particularly of early-stage, low-grade cancers [4]. However, evolving screening guidelines, aimed at balancing the benefits of early detection against risks of overdiagnosis and overtreatment, have caused fluctuations in these trends over time [5]. Following the 2008 U.S. Preventive Services Task Force (USPSTF) recommendation against routine PSA screening [6], and subsequent American and European guidelines [7,8], several countries observed a decline in PCa incidence [3,9,10], while mortality rates decreased in some countries [11] and stabilized in others after years of continuous decline [3,10].

In Italy, PCa is the most frequently diagnosed cancer among men, accounting for about 20% of all new male cancer cases, with over 40,000 diagnoses estimated in 2024 [12]. Despite its high incidence, mortality rates remain relatively low and have shown a gradual decline over the past two decades [3]. PSA testing in Italy is not part of an organized national screening program; instead, it is conducted on an opportunistic basis, typically initiated by general practitioners or urologists. The test is covered by the National Health Service, but in the absence of formal invitation protocols, uptake varies considerably across regions [13]. In Friuli Venezia Giulia (FVG), a region in northeastern Italy with approximately 1.2 million inhabitants (586,000 men in 2020), PSA use began to rise in the early 1990s, contributing to a sharp increase in PCa incidence [9]. To date, screening remains opportunistic in FVG as well, with no structured population-based approach.

Beyond affecting incidence, shifts in screening policies have also influenced cancer classification at diagnosis, particularly as reflected in the Gleason score (GS). This key histopathological grading system evaluates PCa aggressiveness and serves as a strong prognostic marker. The phenomenon of stage migration, where increased early detection results in a higher proportion of localized cases, has significantly altered the distribution of GS. Eventually, changes in detection dynamics, along with refinements in grading criteria, have contributed to a decline of low-grade cancers (GS 2–6) and a relative increase in high-grade cases (GS 8–10) [14]. These patterns highlight the complex interplay between evolving screening practices, shifting disease classification, and advancements in treatment strategies.

Despite the well-documented impact of PSA screening on PCa epidemiology, studies reporting incidence and survival trends by GS remain scarce [15]. Given ongoing changes in detection and management, updated epidemiological data are essential to assess long-term effects on cancer characteristics and outcomes.

This study aims to analyze long-term trends in PCa incidence and survival, focusing on variations by GS and age, using data from the population-based FVG Cancer Registry.

## 2. Materials and Methods

This retrospective, observational, population-based study included all PCa cases diagnosed between 2000 and 2020 among male residents of FVG. Data were obtained from the FVG Cancer Registry, which has systematically recorded all incident cancer cases since 1995, following the guidelines of the International Agency for Research on Cancer (IARC) [16] and the recommendations of the European Network of Cancer Registries (ENCR) [17]. The FVG Cancer Registry ensures high data validity, accuracy, and completeness, consistent with international quality standards [18,19,20].

For the aims of this analysis, the starting year 2000 was chosen to ensure the availability of GS data. PCa diagnoses identified only at autopsy were excluded.

PCa cases were classified into three GS categories based on histopathological reports at diagnosis: low-grade (GS 2–6), intermediate grade (GS 7), and high-grade (GS 8–10). The GS reflects the degree of glandular differentiation in prostate adenocarcinoma and is calculated by summing the two most predominant histologic patterns, graded from 1 (well-differentiated) to 5 (poorly differentiated), with total scores ranging from 2 to 10. Higher GS values indicate more aggressive cancer behavior and a worse prognosis [21].

Cases were further stratified into three age groups: <65, 65–74, and ≥75 years. These categories were chosen for their clinical and epidemiological relevance in PCa research. The <65 group captures early-onset PCa, which may present distinct biological and clinical features. The age groups 65–74 and ≥75 years reflect the age distribution of PCa cases (median: 70 years; interquartile range: 65–76), facilitating comparisons with previous studies. The ≥75 group is specifically emphasized due to clinical recommendations that advocate for age-tailored diagnostic and treatment strategies in older patients.

Age-standardized incidence and mortality rates (per 100,000 person-years) were calculated using the direct method and the 2013 European Standard Population as reference. Mortality data, available from 2006 to 2020, were obtained from the Italian National Institute of Statistics (ISTAT) to complement the incidence analysis. All computations were performed using SEER*Stat software [22].

Temporal trends in incidence—overall and stratified by GS and age—were evaluated using Joinpoint regression analysis to detect significant changes over time. Annual Percentage Change (APC) and corresponding 95% confidence intervals (CIs) were estimated by fitting a log-linear regression model to the data, assuming a constant rate of change over each time segment. The Joinpoint Regression Program was used [23], allowing for the maximum number of joinpoints based on the number of observation points. Trends were classified as increasing or decreasing if the APC was statistically significant; otherwise, they were classified as stable [24]. A two-sided t-test with a significance level of 0.05 was used to determine whether the APC differed significantly from zero.

Trends in the proportion of cases by GS and age were also assessed using the same joinpoint approach to detect changes in cancer grade over time.

Survival analyses included overall survival (OS) and net survival (NS), with NS estimated using the cohort approach and the Pohar Perme method to control for competing deaths and allow comparison over time and across populations [25,26]. Five-year survival was stratified by GS and age to assess prognostic differences.

## 3. Results

A total of 21,571 PCa cases were analyzed, with GS data available for 19,232 cases (Table 1). Among these, 7821 (40.7%) were classified as GS 2–6; 7115 (37.0%) as GS 7; and 4296 (22.3%) as GS 8–10. Patients aged < 65 years had a higher proportion of GS 2–6 (45.2%), while those aged ≥ 75 years exhibited the highest proportion of GS 8–10 cancers (30.0%). From 2000 to 2004, and from 2015 to 2020, GS 2–6 cases declined from 56.9% to 28.4%, while GS 8–10 cases rose from 18.9% to 27.4%.

PCa incidence, NS, and mortality trends are shown in Figure 1. Incidence increased from 2000 to 2007 (APC = +1.8%; 95% CI: +0.5; +4.7) (Figure 1A), then sharply declined from 2007 to 2010 (APC = −7.6%; 95% CI: −9.8; −3.5). Between 2010 and 2020, incidence showed a non-significant decrease (APC = −2.2%; 95% CI: −2.9; +0.7). Concurrently, the 1-year NS increased from 97.0% in 2000–2004 to 98.9% in 2015–2019 (Figure 1B); the 5-year NS rose from 90.0% in 2000–2004 to 95.9% in 2005–2009, stabilizing around 95% in the subsequent periods; and the 10-year NS improved from 84.3% in 2000–2004 to 92.5% in 2010–2014. Mortality rates (Figure 1C) steadily declined from 2006 to 2020, with an APC of −2.2% (95% CI: −3.5; −0.8).

Figure 2 illustrates time trends in the incidence and proportions of PCa cases by GS. The incidence of GS 2–6 cancers showed a significant decline over time (APC = −5.9%; 95% CI: −7.1%; −4.9%) (Figure 2A), while GS 7 cancers initially increased until 2007 (APC = +12.0%; 95% CI: +9.1%; +15.7%) and stabilized thereafter. GS 8–10 incidence remained stable throughout 2000–2020 (APC = +0.2%; 95% CI: −0.7%; +1.2%). However, their proportion (Figure 2B) grew steadily from 20% in 2000 to 29% in 2020, with a notable rise from 2002 (APC = +3.1%; 95% CI: +2.3%; +4.8%).

Trends by age group (Figure 3) showed a general decline in GS 2–6 incidence across all ages, with the most pronounced reduction observed in men aged ≥ 75 years (APC = −8.1%; 95% CI: −9.3%; −7.1%) (Figure 3C). In men aged < 65 years (Figure 3A), incidence showed a non-significant increase from 2000 to 2002 (APC = +26.3%; 95% CI: −3.6%; +63.1%), followed by a decline thereafter (APC = −6.1%, 95% CI: −12.1%; −5.0%). This was reflected in a consistent decrease in the proportion of low-grade cancers over time, across all age groups (Appendix A).

For GS 7 cancers, a rise in incidence was observed in younger men (Figure 3A) until 2006 (APC = +13.6%; 95% CI: +7.7%; +23.5%), followed by a non-significant decline until 2018 (APC = −3.3%; 95% CI: −9.8%; +10.0%). A similar pattern emerged in men aged 65–74 years (Figure 3B), with a sharp increase until 2006 (APC = +14.0%; 95% CI: +7.9%; +32.3%), after which the trend stabilized. In men aged ≥ 75 years (Figure 3C), G7 incidence fluctuated, with alternating periods of non-significant decline and increase. Overall, the GS 7 proportion increased in all ages (Appendix A).

Regarding GS 8–10 cancers, the incidence remained relatively stable across all age groups, with only a modest increase in men aged 65–74 (APC = +1.0%; 95% CI: +0.1%; +1.9%) (Figure 3B). However, the proportion of these high-grade cancers increased steadily among men aged ≥ 65 years, with the most marked rise observed in those aged ≥ 75 years (APC = +3.0%; 95% CI: +2.1%; +4.2%) (Appendix A).

Five-year OS and NS by GS are presented in Figure 4. OS improved for GS 2–6 from 81.4% in 2000–2004 to 88.2% in 2015–2019, and for GS 7 from 78.1% to 88.1%. NS remained high, particularly for GS 2–6 (98.9% to 99.8%) and GS 7 (92.8% to 99.9%). High-grade cancers showed gains in OS (62.9% to 68.0%) and NS (77.9% to 81.9%).

For patients aged < 75 years (Figure 5), OS for GS 2–6 increased from 87.8% in 2000–2004 to 92.5% in 2015–2019, and for GS 7 from 86.8% to 92.0%, with smaller gains for GS 8–10 (73.9% to 77.9%). NS remained above 97% for GS 2–6 and GS 7, reaching 100% from 2015 to 2019. In patients aged ≥ 75 years, OS was lower but improved: for GS 7, from 51.9% in 2000–2004 to 78.1% in 2015–2019, and for GS 8–10 from 43.9% to 54.4%. NS slightly declined for GS 2–6, reaching 97% in 2015–2019, while significant increases were observed after 2005 for GS 7 (NS > 99% vs. 79.6% in 2000–2004) and GS 8–10 (NS > 75% vs. 68.4% in 2000–2004).

## 4. Discussion

This population-based study provides updated figures on PCa incidence and survival trends in the FVG region, focusing on GS variations over time. A marked decrease in the incidence of low-grade cancers (GS 2–6) was documented, while the incidence of high-grade cancers (GS 8–10) remained stable. However, the proportion of high-grade cancers gradually increased, particularly in older age groups. Moreover, survival improved significantly across all GS groups, with particularly substantial gains among individuals aged ≥75 years, although still lower compared to younger men.

Our findings are consistent with previous investigations [3,11,27], showing an initial rise in PCa incidence linked to widespread PSA testing, followed by a decline after screening recommendations became more selective. Similar trends have been observed in most European countries [3,11], where variations in screening intensity have been linked to fluctuations in incidence rates. While PSA testing has contributed to increased incidence, its effect on mortality remains uncertain, as mortality declined even in settings with reduced testing [3]. The mild and steady decline in mortality observed in FVG mirrors national [10] and international patterns [11,28], and could be attributed to the use of PSA testing, as well as advances in effective treatments for late-stage PCa.

As in other studies [15,29], our results highlight a consistent decline in the incidence of low-grade cancers, which are typically slow-growing and associated with a favorable prognosis (i.e., NS approximately 100%). This trend is consistent with increasingly selective screening practices aimed at limiting overdiagnosis and overtreatment of indolent cancers [6,30], in line with guidelines discouraging PSA screening in low-risk individuals. Notably, this trend varied by age: in younger men, incidence initially increased and subsequently declined; in older men, the decline was more linear and pronounced throughout the study period. This age-specific pattern may reflect an initial focus of PSA testing on younger populations, leading to a temporary rise in indolent cancer diagnoses, followed by a shift toward more conservative screening policies, particularly in older age groups. These findings highlight the impact of age-specific screening policies on cancer incidence and support ongoing efforts to tailor PSA testing based on age and individual risk.

Intermediate-risk cancers showed more variable incidence patterns, with an initial increase followed by stabilization or a slight decline in some age groups. GS 7 includes both GS 3 + 4 and GS 4 + 3 cancers, which differ in prognosis and may be differentially impacted by clinical decision-making [21,31]. Our preliminary stratified analyses indicated that GS 3 + 4 and GS 4 + 3 followed broadly similar temporal patterns in terms of incidence and survival. However, further disaggregation would have considerably reduced the robustness of the trend estimates, especially in age-specific analyses. Therefore, we retained GS 7 as a single category, and this internal heterogeneity should be considered when interpreting the results. The observed increasing proportion of GS 7 cancers across all ages may reflect improved diagnostic precision, evolving treatment strategies, and growing awareness of intermediate-risk disease. These trends underscore the importance of careful risk stratification and may inform future updates to active surveillance criteria and treatment guidelines in this heterogeneous group.

Although the incidence of high-grade cancers remained stable, their proportion increased over time, indicating that aggressive cancers now account for a larger share of all diagnoses. This pattern was particularly evident among men aged ≥ 65 years, possibly reflecting both the natural progression of the disease and reduced low-grade detection due to restricted PSA use in older populations. Indeed, some studies have suggested that reduced PSA screening following the USPSTF guidelines may have contributed to a shift toward diagnosing more advanced and metastatic PCa [32]. Our findings are in line with previous reports showing rising proportions of high-risk cancers in the post-PSA era. A US study (2004–2014) [15] reported a stable incidence of higher-grade cancer overall, with a slight decline in men aged ≥ 75 years. In this age group, the proportion of GS 2–6 decreased sharply, while the proportion of aggressive cancers increased after 2010. Another study (1996–2019) found that the percentage of patients with GS ≥ 6 significantly increased between 2016 and 2019 compared to earlier periods (1996–2006 and 2007–2015). This increase was particularly evident in patients aged < 80 years [33].

Survival in PCa has improved over time but remains strongly influenced by cancer characteristics, particularly the GS, with higher grades generally associated with poorer prognosis despite treatment advances [34]. Our analysis showed significant improvements in both OS and NS across all GS categories, most notably between 2000–2004 and subsequent periods. While survival for low-grade cancers was already high, further improvements suggest the benefit of ongoing advancements in diagnostic and management strategies. For high-risk cancers, despite lower survival, the improvements observed may be attributable to more effective therapies targeting aggressive disease. A recent study also reported better outcomes for GS 9–10 cancers over the past two decades [35]. Notably, survival gains in our analysis were particularly marked for GS ≥ 7 cancers in patients aged ≥ 75 years, pointing to better treatment access and efficacy in older populations. This aligns with other studies reporting high survival rates in elderly patients with aggressive disease [36]. A French population-based registry also documented rising NS over time in men aged 60–74 and ≥75 years, with 8-year NS exceeding 70%, though high-grade cancer survival in older men remained around 51% [37]. Overall, our findings point to a combination of enhanced detection of clinically relevant cancers and advancements in treatment strategies as key drivers of survival gains, although changes in grading practices may have contributed to these trends [35].

Early-onset PCa (diagnosed at <65 years) deserves special attention, as it represents a distinct clinical and epidemiological entity. Although these patients account for a minority of cases, they often present with biologically more aggressive features despite their younger age. They are characterized by longer life expectancy, which can affect both treatment choices and long-term prognosis [38]. Although survival was overall better in younger patients, accurate risk stratification remains crucial to ensure appropriate management [39]. A deeper understanding of incidence and outcome patterns in this subgroup is essential to support personalized care and to inform future screening and treatment strategies.

Some limitations of this study should be acknowledged. Firstly, the absence of clinical stage data (i.e., extent of disease) limits our ability to fully interpret incidence and survival trends. Furthermore, the lack of information on individual treatments introduces potential confounding, as advances in hormonal therapies and radiotherapy may have improved outcomes independently of cancer characteristics. Similarly, we were unable to consider patient comorbidities, which can strongly influence treatment decisions and outcomes, particularly in older populations.

Improvements in imaging modalities and diagnostic workups over the study period may have led to stage migration, potentially impacting trends.

Approximately 11% of cases had missing GS data; however, the distribution of missing values appeared balanced across years and groups, suggesting a limited risk of bias. Finally, the classification of GS may have been affected by temporal changes in pathology reporting criteria [21], which could impact the comparability of GS categories across the study period.

Despite these limitations, the study benefits from the use of high-quality, population-based cancer registry data, ensuring comprehensive case ascertainment and minimizing selection bias. Moreover, the extended study period (2000–2020) allowed for the assessment of long-term trends in PCa incidence and survival stratified by GS.

## 5. Conclusions

This study highlights significant variations in PCa incidence and survival by GS over two decades. The decline in the incidence of low-grade cancers and the stable rates of high-grade cancers reflect evolving screening and diagnostic practices. However, the increasing proportion of aggressive disease in older populations warrants targeted clinical attention and highlights the need to address persistent challenges in managing these cases. At the same time, the observed improvements in survival, particularly in those aged ≥75 years, underscore the positive impact of advances in therapeutic strategies. Continued monitoring of screening protocols and treatment approaches remains essential to ensure further progress in managing both low- and high-risk PCa, while minimizing the harms of overdiagnosis in future screening programs. Future research should explore clinical outcomes by disease stage and treatment to better inform patient-centered strategies.

## Figures and Tables

**Figure 1 curroncol-32-00426-f001:**
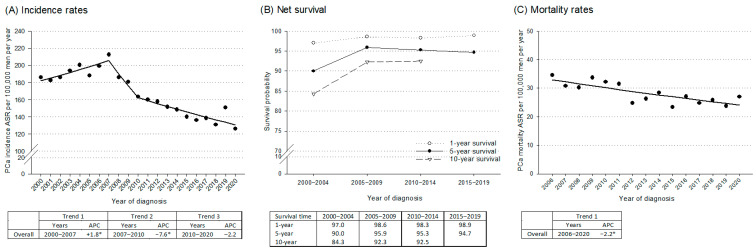
Trends in prostate cancer (PCa) incidence, net survival, and mortality over time: (**A**) Age-standardized incidence rates ^a^ (ASR) of PCa per 100,000 men per year with corresponding joinpoint analyses; (**B**) 1-, 5-, and 10-year net survival probabilities; (**C**) Age-standardized mortality rates ^a^ (ASR) of PCa per 100,000 men per year with corresponding joinpoint analyses. Friuli Venezia Giulia, Italy, 2000–2020. In panels (**A**,**C**), the data points represent observed ASRs, while the lines illustrate the trends modeled by joinpoint regression analysis. For each segment, the Annual Percent Change (APC) is calculated and presented in the accompanying table, with the respective trend segments identified. ^a^ Age-standardized to the 2013 European Standard Population. * *p* < 0.05.

**Figure 2 curroncol-32-00426-f002:**
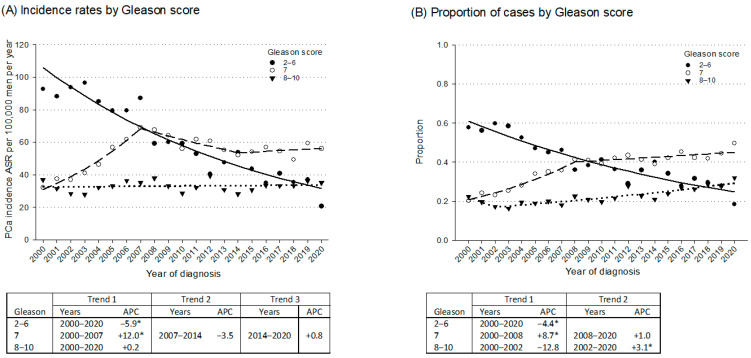
Trends in age-standardized incidence rates (ASRs) ^a^ and proportion of prostate cancer (PCa) cases by Gleason score, with corresponding joinpoint analyses: (**A**) Incidence rates by Gleason score, all ages; (**B**) Proportion of cases by Gleason score, all ages. Friuli Venezia Giulia, Italy, 2000–2020. The data points represent observed ASRs and proportions of cases, while the lines illustrate the trends modeled by joinpoint regression analysis. For each segment, the Annual Percent Change (APC) is calculated and presented in the accompanying table, with the respective trend segments identified. ^a^ Age-standardized to the 2013 European Standard Population. * *p* < 0.05.

**Figure 3 curroncol-32-00426-f003:**
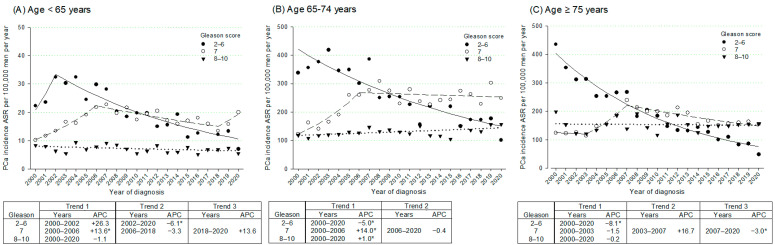
Trends in age-standardized incidence rates (ASR) ^a^ of prostate cancer (PCa) cases by Gleason score and age group, with corresponding joinpoint analyses: (**A**) Age < 65 years; (**B**) Age 65–74 years; (**C**) Age ≥ 75 years. Friuli Venezia Giulia, Italy, 2000–2020. The data points represent observed ASRs, while the lines illustrate the trends modeled by joinpoint regression analysis. For each segment, the Annual Percent Change (APC) is calculated and presented in the accompanying table, with the respective trend segments identified. ^a^ Age-standardized to the 2013 European Standard Population. * *p* < 0.05.

**Figure 4 curroncol-32-00426-f004:**
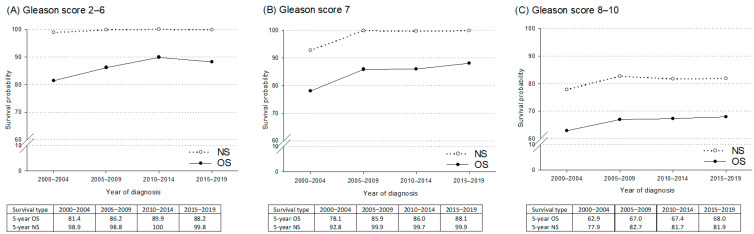
Five-year observed survival (OS) and net survival (NS) probabilities for prostate cancer cases by Gleason score: (**A**) Gleason 2–6; (**B**) Gleason 7; (**C**) Gleason 8–10. Friuli Venezia Giulia 2000–2020.

**Figure 5 curroncol-32-00426-f005:**
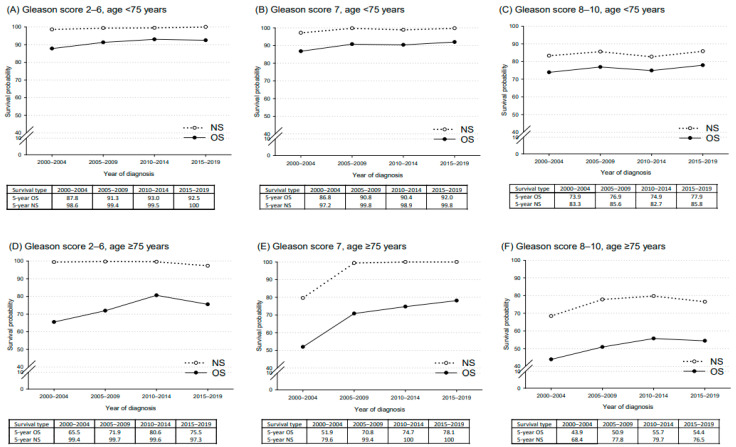
Five-year observed survival (OS) and net survival (NS) probabilities for prostate cancer cases by Gleason score and age group: (**A**) Age < 75 years, Gleason 2–6; (**B**) Age < 75 years, Gleason 7; (**C**) Age < 75 years, Gleason 8–10; (**D**) Age ≥ 75 years, Gleason 2–6; (**E**) Age ≥ 75 years, Gleason 7; (**F**) Age ≥ 75 years, Gleason 8–10. Friuli Venezia Giulia 2000–2020.

**Table 1 curroncol-32-00426-t001:** Distribution of 21,571 prostate cancer (PCa) cases by Gleason score and selected variables. Friuli Venezia Giulia, Italy, 2000–2020.

	Total PCa Cases	PCa Cases Missing Gleason Score	PCa Cases with Gleason Score	Gleason Score		
				2–6	7	8–10
	No.	No. (row%)	No.	No. (row%)	No. (row%)	No. (row%)
All	21,571	2339 (10.8)	19,232	7821 (40.7)	7115 (37.0)	4296 (22.3)
Age at diagnosis						
<65	5253	394 (7.5)	4859	2194 (45.2)	1888 (38.8)	777 (16.0)
65–74	9553	828 (8.7)	8725	3558 (40.8)	3341 (38.3)	1826 (20.9)
≥75	6765	1117 (16.5)	5648	2069 (36.6)	1886 (33.4)	1693 (30.0)
Period of diagnosis						
2000–2004	5127 (~1025/year)	692 (13.5)	4435	2522 (56.9)	1074 (24.2)	839 (18.9)
2005–2009	5760 (~1152/year)	600 (10.4)	5160	2200 (42.6)	1926 (37.3)	1034 (20.1)
2010–2014	5014 (~1003/year)	510 (10.2)	4504	1643 (36.5)	1845 (41.0)	1016 (22.5)
2015–2020	5670 (~945/year)	537 (9.5)	5133	1456 (28.4)	2270 (44.2)	1407 (27.4)

## Data Availability

Data are unavailable due to privacy restrictions.

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
