# Peer review of "Trends in Prostate Cancer Incidence and Survival by Gleason Score from 2000 to 2020: A Population-Based Study in Northeastern Italy"

_curroncol, 2025, doi:10.3390/curroncol32080426_

Round 1
Reviewer 1 Report
Comments and Suggestions for Authors
Thank you for the opportunity to review the manuscript ID: cancers-3749934. This manuscript aimed to assess the long-term trends in prostate cancer incidence and survival by Gleason Score in Friuli Venezia Giulia, northeastern Italy.
Comments:
Section Introduction:
Line 67:
Add a new paragraph describing the prostate cancer burden of disease in Italy. Cite appropriate references.
Add a new paragraph in which the practice of prostate cancer screening in Italy should be presented (including when it was implemented, whether it is mandatory or not, whether it is paid or free, whether it is organized-national or regional only, which population is eligible for prostate cancer screening, what is the coverage of prostate cancer screening, etc.).
In particular, describe possible changes in the characteristics of the prostate cancer screening program during the period relevant to the incidence and mortality presented in this study.
For all of the above questions, indicate the specifics of screening for prostate cancer in Friuli Venezia Giulia, northeastern Italy.
Cite appropriate references.
Section Methods:
Line 85: Specify `Study design`.
Line 92: Specify `Data Quality` (including validity, accuracy, completeness of the data presented in this paper, etc.), citing appropriate references.
Lines 93-94:
Define Gleason Score, citing appropriate references.
List all measures presented in this manuscript.
Line 95: State the rationale for this age distribution in this work.
Lines 97-99: State the standardization method used in this study.
Lines 101-104:
State the significance test used to assess the differences.
Specify the minimum and maximum number of joinpoints selected for the joinpoints regression analysis in this study.
Define the meaning of the terms `increase`, `decrease and `stable`, which are used in this paper to describe the changes of the trends, according to the results of the joinpoint regression analysis. Cite appropriate references.
Show precise results of comparison by age groups and by Gleason Scores, using the comparability test.
Conduct sensitivity analysis in order to increase the external validity of this paper.
Line 111: Delete word `Supplementary`.
Line 115: List the average age of the subjects in this study, as well as the minimum and maximum ages.
Lines 153-165: Match the type of rates shown in Figure 3 with the description in the corresponding text.
Lines 166-173: Check and match the description of trends presented in this text with the definitions of the meaning of the terms `increase`, `decrease` and `stable`, which in this paper describe the trends in the Methods section, according to the results of the applied joinpoint regression analysis.
Line 269: Add a new paragraph in which the contribution and importance of early-onset prostate cancer in the trends shown in this study should be stated.
Lines 270-274: Supplement this text in this paragraph with a discussion of the numerous other limitations of this paper.
Reviewer 2 Report
Comments and Suggestions for Authors
The manuscript addresses an important and clinically relevant topic. The study leverages robust population-based cancer registry data over 20 years, employs appropriate statistical tools, and provides insightful public health and clinical implications related to changing screening practices and therapeutic advances. However, there are still some issues that require clarification before publication:
- Although the abstract is generally informative, it is quite dense. Consider simplifying the language to improve accessibility for a broader audience. Also, ensure that all major findings—such as the stable incidence yet increasing proportion of GS 8–10 tumors—are more clearly emphasized. The Simple Summary could benefit from more concise phrasing and a clearer focus on the clinical significance of the findings for older patients.
- The study groups GS into three main categories (2–6, 7, 8–10), but GS 7 is heterogeneous (i.e., GS 3+4 vs. 4+3). It would be helpful to discuss whether this internal heterogeneity was considered and how it might influence the interpretation of trends. If not feasible to re-analyze, this limitation should be explicitly acknowledged in the Discussion.
- The manuscript mentions a high-quality cancer registry, but some variables (GS missing in 10.8% of cases) could bias results. A discussion of how missing data were handled—especially whether they are randomly distributed or concentrated in certain subgroups—should be added to enhance transparency.
- The discussion acknowledges the lack of clinical stage and possible changes in pathology grading criteria. However, the potential confounding effects of treatment changes over time, patient comorbidities, or improvements in imaging techniques are not addressed. Including these would strengthen the manuscript’s scientific rigor.
- The Discussion section is informative, but could be more focused. Some sections are overly descriptive and repeat results. The authors should aim to reduce redundancy, enhance interpretive commentary, and more clearly relate findings to clinical and public health decision-making, particularly regarding age-specific screening guidelines.
- The manuscript would benefit from a thorough language revision. Several long and complex sentences hinder readability. Employing a professional editing service or a native English-speaking reviewer is strongly recommended to improve sentence flow and clarity.
The manuscript would benefit from a thorough language revision. Several long and complex sentences hinder readability. Employing a professional editing service or a native English-speaking reviewer is strongly recommended to improve sentence flow and clarity.
Round 2
Reviewer 1 Report
Comments and Suggestions for Authors
Thank you for the opportunity to re-review the manuscript ID: curroncol-3749934.
The authors addressed all my comments, and made appropriate corrections and additions in the revised version of this manuscript.
I believe that this paper is now more informative, as well as a valuable source of information for anyone dealing with this topic.
Thanks to the authors for their efforts in revising this manuscript.